# Longitudinal association between food frequency and changes in body mass index: a prospective cohort study

Antonio Bernabe-Ortiz ,[1,2] Rodrigo M Carrillo-Larco [1,3,4]

¹CRONICAS Centre of Excellence in Chronic Diseases, Universidad Peruana Cayetano Heredia, Lima, Peru
²Universidad Científica del Sur, Lima, Peru
³Department of Epidemiology and Biostatistics, School of Public Health, Imperial College London, London, UK
⁴Centro de Estudios de Población, Universidad Católica los Ángeles de Chimbote (ULADECH-Católica), Chimbote, Peru

**Correspondence to**
Dr Antonio Bernabe-Ortiz;
antonio.bernabe@upch.pe

## ABSTRACT

**Objectives** Analysing data of the Young Lives Study in Peru, we aimed at assessing the association between daily food frequency and body mass index (BMI) changes between 2006 and 2016.

**Design** Secondary analysis of a prospective ongoing cohort study.

**Setting** 20 sentinel sites in Peru.

**Participants** Children enrolled in the younger cohort of the Young Lives Study. We used information from the second (2006–2007), third (2009–2010), fourth (2013–2014) and fifth (2016–2017) rounds of the younger cohort in Peru.

**Primary and secondary outcomes** BMI as well as BMI-for-age z-score, both as numerical variables.

**Results** Data from 1948 children, mean age 4.3 (SD: 0.3) years and 966 (49.6%) women were included at baseline. In multivariable model, lower food consumption frequency was associated with increased BMI and BMI-for-age z-scores: children reporting <4 times of food consumption per day had a greater increase in BMI (β=0.39; 95% CI 0.17 to 0.62) and BMI-for-age z-score (β=0.07; 95% CI 0.01 to 0.13) compared with those reporting 5 per day. Results were consistent for those reporting exactly eating 4 times per day (β for BMI=0.16; 95% CI 0.02 to 0.30 and β for BMI-for-age z-score=0.05; 95% CI 0.01 to 0.09).

**Conclusions** Children who eat <5 times per day, gain more BMI compared with those who eat ≥5 times. Parents should receive information to secure adequate nutrition for their children, both in terms of quality and quantity.

## INTRODUCTION

Current trends in children's and adolescent's body mass index (BMI) show a sustained and fast growth in different low/middle-income countries (LMICs).[1] Moreover, the swift transition from underweight to overweight and obesity has also been reported in specific countries in Latin America, where the median of the prevalence distribution of excess body-weight among school-aged children has been estimated in 29.8%, while this number was 17.8% among adolescents.[2] Similar figures have been found in Peru among children between 6 and 9 years old.[3]

Changes in BMI are explained by a misbalance between food intake and energy

### Strengths and limitations of this study

► Results are supported by a population-based cohort of children with more than 9 years of follow-up in a middle-income country.
► Our analysis took into account the repeated-measure nature of the data including changes of both outcome and exposure over time.
► Recall bias can be an issue as mother's reports were used to evaluate the number of times of food consumption per day.

expenditure[4]; therefore, some diet characteristics, such as diet quality and excess energy intake, may affect these changes. In that sense, food frequency has been described as a potential risk factor for BMI increase and, for instance, overweight and obesity.[5]

Some studies have reported that regular (ie, more frequent) food frequency is associated with lower risk of overweight and obesity,[6–8] whereas other studies have not reported such association.[9 10] Moreover, a relatively recent systematic review and meta-analysis of 57 observational studies reported a significant relationship between frequent meals and better nutrition outcomes, including lower BMI.[11] Nevertheless, this meta-analysis included only reports from high-income countries, showed correlations because most of the summarised studies were cross-sectional (only eight studies were longitudinal), and only one study was from Latin America (Puerto Rico).[12] Moreover, among those longitudinal studies, the variation of food frequency and BMI over time was not taken into account. Therefore, whether the association between food frequency and BMI holds in Latin America, is unknown; though there could be differences because of the nutritional profiles and access to food patterns of these populations. For example, food insecurity is high in Peru, especially in rural and poor areas,[13] and households with an insecure profile usually have a poor diet

quality and consume more carbohydrates.[14] Thus, this study aimed at assessing the association between food frequency and BMI changes over time among children in Peru. We hypothesised that children eating less frequently will have a greater gain of BMI over time.

## METHODS
### Study design
This is a secondary analysis using information of the Young Lives Study, a longitudinal ongoing prospective cohort conducted in four LMICs (Ethiopia, India, Peru and Vietnam).[15] The cohort started in 2002 and currently has five assessment rounds, that is, a baseline and four follow-up evaluations.

### Study participants
The Young Lives Study in Peru is comprised by two different cohorts: the younger cohort that initially enrolled 2052 children aged between 6 and 18 months, and the older cohort that included 1000 children between 7 and 8 years.[15] For this work, we used information from the second (2006–2007), third (2009–2010), fourth (2013–2014) and fifth (2016–2017) rounds of the younger cohort in Peru as information about food frequency was available in that cohort. As a result, information of the second round was used as the baseline of the cohort, and information from the third, fourth and fifth rounds were included as follow-up assessments. Records with incomplete information in the variables of interest (food frequency and BMI) were excluded from the analysis.

### Sampling
A sentinel site sampling strategy was used as previously published.[16] Briefly, a multistage, cluster stratified, random sampling technique of sentinel sites was conducted. The initial sample frame was carried out at the district level, selecting 20 sentinel sites from a total of 1818 districts available. To oversampling poor areas, the top 5% richest districts were not included in the sampling process.

Maps of census tracts, comprising block of houses or set of houses, were used to randomly select one census tract in each district using a random number table. All households in any given block or set of houses were visited to identify one household with at least one child meeting inclusion criteria. Different block or set of houses were approached until the total eligible households were completed. Three different teams, including fieldworkers, data-entry staff and supervisors were responsible for data collection and quality of six or seven sentinel sites.

### Study variables
Two were the outcomes of interest for this study: BMI evaluated as numerical continuous variable and BMI-for-age z-score. Both height and weight were measured in each children assessment and used to estimate BMI as usual. BMI-for-age z-score was based on the WHO reference.[17]

We used these outcomes because we aimed at assessing changes over time using the information available from all the four study rounds. In addition, both definitions (ie, BMI and BMI-for-age z-score) were used as BMI alone is not the best indicator of excess of weight among children under 6 years of age.[17 18]

The exposure of interest was food frequency as reported by the parents based on seven questions related to the consumption of any possible meal or snack during the previous 24 hours before the interview. The question was asked as follow: during the previous 24 hours period, did the child consume: any food before a morning meal (breakfast)? Any morning meal (breakfast)? Any food between morning and midday meals? Any midday meal? Any food between midday and evening meals? Any evening meal? Any food after the evening meal? A simple addition of yes/no responses to these seven questions was used to estimate the total food frequency the child ate in the last 24 hours. For analysis purposes, the variable was categorised as <4, 4, 5, 6 and 7 times per day,[5] and the middle category (5 times per day) was used as the reference category. This process was conducted for each assessment included in the analysis (ie, baseline and three follow-up evaluations).

Other variables were also included as potential confounders in regression models as evaluated at baseline: sex (boy or girl), age (in years), socioeconomic status assessed using a wealth index created based on household assets and split in tertiles (low, middle and high), setting type (urban or rural), maternal and paternal education based on years of education (<7 years, 7–11 years and ≥12 years), and maternal BMI as continuous variable, but also split according to traditional thresholds (normal ($<25\,kg/m^2$), overweight ($25$–$29\,kg/m^2$) and obese ($\geq30\,kg/m^2$)).[18]

### Data analysis
STATA V.13 for Windows (StataCorp) was used for statistical analysis. Initially, the description of the study population was tabulated according to food frequency using mean and SD for numerical variables and proportions for categorical ones.

$\chi^2$ test or analysis of variance was used for comparison between variables accordingly.

Means and SD of BMI and BMI-for-age z-score for baseline and each follow-up were calculated according to food frequency at baseline (ie, without taking into account correlation between measurements or any other variables). Finally, crude and adjusted linear mixed models with random intercepts were used to determine the association of interest (ie, association between BMI or BMI-for-age z-score and food frequency, accounting for the repeated-measure nature of the study and including changes of both variables over time). Three levels were fitted in the regression models (children assessment as level 1, subject as level 2 and sentinel site as level 3). BMI changes over time were presented as coefficients and their respective 95% CIs controlling for different confounders.

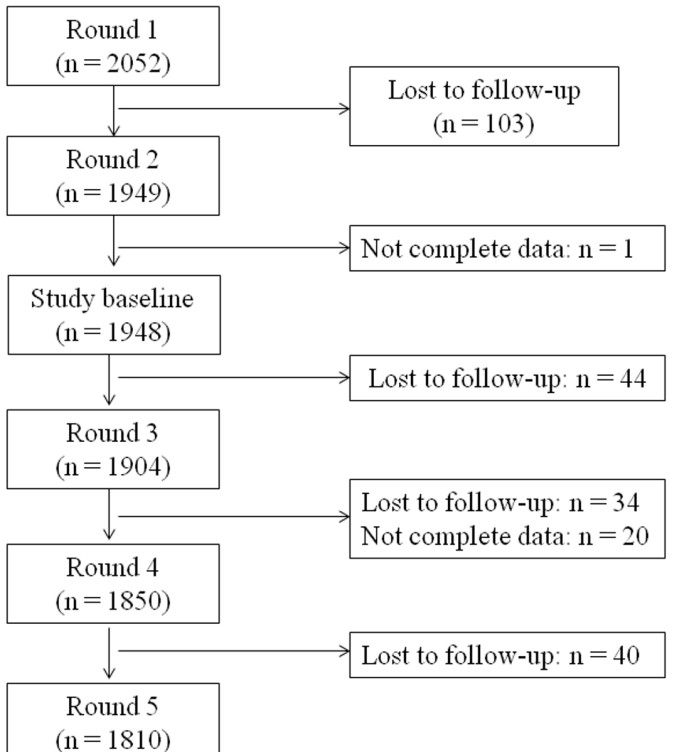

**Figure 1** Flowchart of participants analysed using the young lives younger cohort.

## RESULTS

### Baseline characteristics of the study population

A total of 2052 children were originally recruited in the younger cohort in Peru. Of them, 103 (5.0%) were lost to follow-up and 1 (0.1%) was excluded because of missing data (figure 1). Thus, 1948 individuals were included in the baseline analysis, with a mean age of 4.3 (SD: 0.3) years and 966 (49.6%) were women. At baseline, food consumption frequency was on average 4.9 (SD: 1.0, range: 2–7) times per day and 847 (43.5%) reported eating 5 times per day, whereas 215 (11.0%) ate <4 times per day and 77 (4.0%) ate ≥7 times per day. The characteristics of the study population according to food frequency at baseline are shown in table 1. Of note, children age, maternal and paternal education, socioeconomic status, setting type and maternal BMI were positively associated with food frequency.

### Food frequency and BMI during follow-up

Figure 1 shows in detail the number of participants in each of the follow-up assessments. On average, children were followed up for a total of 9.6 (SD: 0.3) years. The proportion of children that reported eating ≥5 times per day dropped from 68.4% at baseline (4.3±0.3 years of age) to 49.8% at the end of the follow-up (13.9±0.1 years of age).

Table 2 shows a description of the mean BMI during each study follow-up rounds without taking into account intra-subject correlation. All the food frequency groups showed an increase of BMI and BMI-for-age z-score, and this increase seemed to be higher in the group reporting a food frequency of 5 times per day.

### BMI and BMI-for-age z-score change over time

Results of the crude and adjusted regression models are shown in table 3. According to multivariable model, our results show that lower food frequency consumption was associated with increased BMI and BMI-for-age z-scores. Children reporting <4 meals per day had a greater increase in BMI (β=0.39; 95% CI 0.17 to 0.62) and BMI-for-age z-score (β=0.07; 95% CI 0.01 to 0.13) compared with those reporting eating 5 times per day. These results were similar for those reporting eating exactly 4 times per day (β for BMI=0.16; 95% CI 0.02 to 0.30 and β for BMI-for-age z-score=0.05; 95% CI 0.01 to 0.09); however, changes among those eating 6 or 7 times per day did not show association (see figure 2).

## DISCUSSION

### Main findings

Our study shows evidence of an association between daily food frequency and children's BMI changes over time. Although the magnitude of the association was small, less frequent food consumption was associated with greater BMI gain over time: those eating less than 5 times per day had greater BMI and BMI-for-age z-score measurements, whereas those eating ≥5 times per day did not have an increase in these outcomes. In addition, almost two-thirds of the study population reported eating ≥5 times per day at baseline (ie, when the child aged between 4 and 5 years), but this number reduced to less than half of participants when children were between 14 and 15 years.

### Comparison with previous studies

Two systematic reviews have approximated to the association proposed in this paper. The first one reported inconsistent and weak evidence of an inverse association between food frequency and risk of childhood overweight.[19] This review used 15 reports and only 4 were longitudinal studies. The second systematic review, conducted in 2018, found a significant relationship between frequent food consumption and better nutritional health using 57 studies, both in younger and older children, across countries and socioeconomic groups.[11] However, the latter review only reported weighted mean correlations and no risk of overnutrition or changes of BMI over time, and many of the summarised reports studied convenience samples. Our work agrees with the current literature and adds more evidence from new populations (children in Latin America as limited information is available) and followed a strong observational design (prospective cohort).

Many of the studies looking for the association of interest have been cross-sectional, and have used outcomes based on overweight or obesity. Toschke *et al* reported a decrease in the prevalence of obesity by

**Table 1** Characteristics of the study population according to food frequency at baseline

| | Food frequency at baseline (times per day) | | | | | |
|---|---|---|---|---|---|---|
| | <4 (n=217) | 4 (n=402) | 5 (n=851) | 6 (n=409) | 7 (n=77) | P value* |
| Sex, n (%) | | | | | | 0.52 |
| Female | 102 (47.0) | 201 (50.0) | 438 (51.5) | 191 (46.7) | 39 (50.7) | |
| Male | 115 (53.0) | 201 (50.0) | 413 (48.5) | 218 (53.3) | 38 (49.4) | |
| Age, mean (SD) | | | | | | <0.001 |
| In years | 4.2 (0.2) | 4.3 (0.3) | 4.4 (0.3) | 4.3 (0.3) | 4.3 (0.2) | |
| Maternal education, n (%) | | | | | | <0.001 |
| <7 years | 161 (74.5) | 219 (54.9) | 312 (37.1) | 150 (36.7) | 28 (36.8) | |
| 7–11 years | 47 (21.8) | 139 (34.8) | 340 (40.4) | 158 (38.6) | 28 (36.8) | |
| 12+ years | 8 (3.7) | 41 (10.3) | 189 (22.5) | 101 (24.7) | 20 (26.3) | |
| Paternal education, n (%) | | | | | | <0.001 |
| <7 years | 116 (56.0) | 149 (38.3) | 248 (30.0) | 112 (28.3) | 24 (32.0) | |
| 7–11 years | 80 (38.7) | 181 (46.5) | 369 (44.6) | 185 (46.7) | 27 (36.0) | |
| 12+ years | 11 (5.3) | 59 (15.2) | 211 (25.4) | 99 (25.0) | 24 (32.0) | |
| Socioeconomic status, n (%) | | | | | | <0.001 |
| Low | 144 (66.3) | 167 (41.6) | 219 (25.7) | 100 (24.5) | 24 (31.2) | |
| Middle | 57 (26.3) | 148 (36.8) | 283 (33.3) | 140 (34.2) | 25 (32.5) | |
| High | 16 (7.4) | 87 (21.6) | 349 (41.0) | 169 (41.3) | 28 (36.4) | |
| Setting type, n (%) | | | | | | <0.001 |
| Urban | 53 (24.4) | 195 (48.5) | 544 (63.9) | 248 (60.6) | 42 (54.6) | |
| Rural | 164 (75.6) | 207 (51.5) | 307 (36.1) | 161 (39.4) | 35 (45.4) | |
| Maternal BMI, mean (SD) | | | | | | 0.02 |
| In kg/m$^2$ | 25.3 (3.3) | 25.9 (4.1) | 26.4 (4.5) | 26.1 (3.8) | 25.8 (3.7) | |
| Maternal BMI, n (%) | | | | | | 0.10 |
| Normal | 108 (52.2) | 175 (46.1) | 343 (42.6) | 171 (43.4) | 33 (46.5) | |
| Overweight | 84 (40.6) | 148 (38.9) | 333 (41.4) | 162 (41.1) | 29 (40.8) | |
| Obese | 15 (7.2) | 57 (15.0) | 129 (16.0) | 61 (15.5) | 9 (12.7) | |

*Comparisons were conducted using $\chi^2$ tests for categorical variables, and analysis of variance for numerical ones.
BMI, body mass index.

number of daily meals in two different studies[5 8]; whereas in a cross-sectional assessment of a Finnish cohort study of adolescents they reported that adolescents who ate five meals per day were at lower risk for general and abdominal obesity as well as hypertriglyceridemia.[6] Thus, our study expands on previous findings by using repeated measurements following appropriate longitudinal analysis to verify the association of interest in a population-based cohort study of children transitioning to adolescents.

Of note, our findings support the fact that the number of food consumed per day seems to decrease from childhood to adolescence. Therefore, children may lose the beneficial effects of eating more often throughout the day, thus increasing the probability of having overnutrition during adolescence.

**Public health relevance**

Our data suggest that a higher number of daily food consumption may reduce BMI over time, reducing potentially the risk of overweight and obesity in children and adolescents. Effects of food frequency on endocrine responses and regulation might explain our findings. Increased food frequency may attenuate postprandial metabolic and endocrine responses to nutrient intake, with the subsequent reduction in insulin secretion.[7 20]

Since there is more evidence that skipping meals, especially breakfast, is associated with increasing risk of overnutrition (ie, overweight and obesity) among children,[21] the promotion of a regular food pattern including at least five meals per day should be guaranteed. In addition, the quality and adequate composition of meals must be also promoted to children and, especially, their parents, families and caregivers.

**Table 2** BMI and BMI z-score by food frequency at baseline: mean and SD during follow-up

| | Mean and SD of BMI and BMI z-score | | | |
| --- | --- | --- | --- | --- |
| | Baseline (n=1948) | First follow-up (n=1904) | Second follow-up (n=1850) | Third follow-up (n=1810) |
| Age | | | | |
| Mean age (SD) | 4.3 (0.3) | 6.9 (0.7) | 10.9 (0.1) | 13.9 (0.1) |
| Food frequency | BMI (in kg/m$^2$): mean (SD) | | | |
| <4 times per day | 16.4 (1.9) | 16.4 (1.7) | 18.5 (2.5) | 21.1 (2.7) |
| 4 times per day | 16.4 (1.5) | 16.8 (2.1) | 19.4 (3.0) | 21.7 (3.3) |
| 5 times per day | 16.5 (1.9) | 17.0 (2.4) | 19.9 (3.4) | 21.8 (3.4) |
| 6 times per day | 16.3 (1.6) | 16.9 (2.4) | 19.8 (3.3) | 21.6 (3.4) |
| 7 times per day | 16.5 (2.4) | 16.8 (2.4) | 19.4 (3.3) | 21.0 (2.9) |
| Food frequency | BMI z-score: mean (SD) | | | |
| <4 times per day | 0.67 (1.06) | 0.29 (0.92) | 0.18 (0.92) | 0.29 (0.81) |
| 4 times per day | 0.68 (0.92) | 0.49 (1.02) | 0.50 (1.02) | 0.44 (0.93) |
| 5 times per day | 0.71 (1.09) | 0.59 (1.08) | 0.64 (1.11) | 0.44 (1.01) |
| 6 times per day | 0.65 (0.98) | 0.55 (1.13) | 0.61 (1.08) | 0.41 (0.99) |
| 7 times per day | 0.72 (1.33) | 0.42 (1.14) | 0.45 (1.15) | 0.26 (0.90) |

BMI, body mass index.

## Strength and limitations

Our findings are supported by a population-based cohort of children with more than 9 years of follow-up in a middle-income country. Moreover, our longitudinal

**Table 3** Food frequency and changes over time of BMI: crude and adjusted linear mixed models

| | Crude model Coefficient (95% CI) | Adjusted model* Coefficient (95% CI) |
| --- | --- | --- |
| Food frequency | Change in BMI (kg/m$^2$) | |
| <4 times per day | **1.18 (0.73 to 1.64)** | **0.39 (0.17 to 0.62)** |
| 4 times per day | **0.71 (0.43 to 0.98)** | **0.16 (0.02 to 0.30)** |
| 5 times per day | 1 (Reference) | 1 (Reference) |
| 6 times per day | **−0.54 (−0.68 to −0.40)** | −0.03 (−0.15 to 0.09) |
| 7 times per day | **−0.73 (−1.26 to −0.21)** | 0.08 (−0.33 to 0.50) |
| Food frequency | Change in BMI z-score | |
| <4 times per day | 0.01 (−0.08 to 0.10) | **0.07 (0.01 to 0.13)** |
| 4 times per day | 0.01 (−0.03 to 0.06) | **0.05 (0.01 to 0.09)** |
| 5 times per day | 1 (Reference) | 1 (Reference) |
| 6 times per day | 0.01 (−0.03 to 0.06) | −0.01 (−0.05 to 0.04) |
| 7 times per day | 0.06 (−0.10 to 0.21) | 0.04 (−0.13 to 0.20) |

*Adjusted for sex, age, maternal education, paternal education, wealth index, setting type and maternal BMI (in categories).
Estimates in bold are statistically significant.
BMI, body mass index.

analysis shows strong association between food frequency and BMI, taking into account the repeated-measurements and including changes of both variables over time. However, this study has limitations that should be highlighted. First, parents' reports were used to evaluate the food frequency consumption per day based on the last day before the interview; recall bias may arise as a concern. In addition, parents may not be aware of eating frequency of children, especially at older ages. Moreover, although questions assessed any meal or snack, portion (ie, a large meal or a single bite) was not considered and, for instance, not evaluated as part of the analysis. Nevertheless, our results are congruent with previous studies.[22–24] Second, other obesity markers, such as abdominal obesity, were not available for data analysis. Third, no all the potential confounders were available to adjust our regression models (eg, physical activity or portion size); some residual confounding may be present. Finally, selection bias could be a concern as 5% of the richest districts were excluded during the sampling process, potentially affecting the broad generalisability of our results. Moreover, in countries undergoing nutrition transition such as Peru, changes in BMI are initially seen in the wealthiest individuals.[25] Nonetheless, our estimates are informative for the general population though should be interpreted cautiously for families with high socioeconomic status.

## CONCLUSION

There is evidence of association between the number of foods consumed per day by a child and increasing

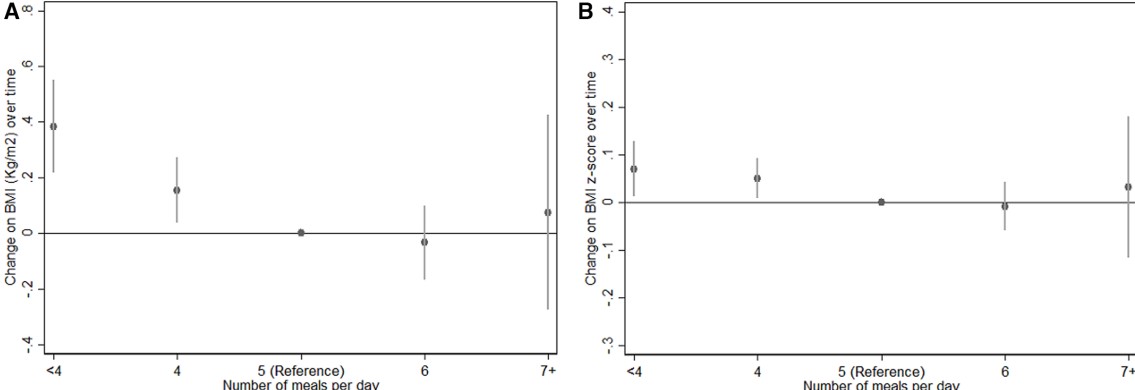

**Figure 2** Variation over time in (A) BMI and (B) BMI-for-age z-score according to food frequency controlling for potential confounders. BMI, body mass index.

BMI over time: those who ate less than 5 times per day had greater BMI and BMI-for-age z-score measurements compared with those eating ≥5 times per day. The promotion of a regular food consumption pattern including at least 5 per day should be guaranteed.

**Contributors** AB-O and RMC-L conceived the idea. RMC-L conducted the data cleaning and harmonisation. AB-O conducted the statistical analysis. AB-O wrote the first version of the manuscript with input from RMC-L. Both authors approved the submitted version.

**Funding** RMC-L is supported by a Wellcome Trust International Training Fellowship (214185/Z/18/Z).

**Competing interests** None declared.

**Patient consent for publication** Not required.

**Ethics approval** The Young Lives study was originally approved by the Ethics Committee Social Science Division, University of Oxford, UK, in 2002. In Peru, the approval was granted by the Research Ethics Committee of the Instituto de Investigacion Nutricional in Lima. Data are freely available under request and information is de-identified.

**Provenance and peer review** Not commissioned; externally peer reviewed.

**Data availability statement** Data are available upon reasonable request. Data are available under request via the Young Lives webpage (https://www.younglives.org.uk/). Statistical code is available at: doi.org/10.6084/m9.figshare.11629092 (STATA syntax).

**ORCID iDs**
Antonio Bernabe-Ortiz http://orcid.org/0000-0002-6834-1376
Rodrigo M Carrillo-Larco http://orcid.org/0000-0002-2090-1856

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
