## [Reviewer comments · BMJ Open]

ARTICLE DETAILS

TITLE (PROVISIONAL)	Longitudinal association between food frequency and changes in body mass index: a prospective cohort study
AUTHORS	Bernabe-Ortiz, Antonio; Carrillo-Larco, Rodrigo

VERSION 1 – REVIEW

REVIEWER	Professor Rachael Taylor University of Otago New Zealand
REVIEW RETURNED	06-Feb-2020

GENERAL COMMENTS	The authors present an interesting analysis of meal frequency in relation to BMI over time in young Peruvian children. Although this question has been examined previously many times, the authors quite rightly highlight that much of the existing evidence is cross-sectional in nature, and in children from high-income countries. I suggest some relatively small edits to the manuscript to improve the clarity of the data: 1. It would perhaps be more useful to have the mean age of participants at each time point rather than the years of data collection.2. Page 6, line 13 - reword to illustrate this more clearly - do the authors refer to the median over different ages? It is not entirely clear what this sentence is referring to.3. The mean (SD) age at each follow-up needs to be included - perhaps in table 2? This would make the BMI data more useful - at present BMI itself is not particularly useful without knowledge of the relevant age of the children, and how variable that age was since BMI changes so rapidly across growth.4. What reference data were used to calculate the BMI z-scores?5. More information - and acknowledgement of the limitations - regarding the meal frequency measure needs to be provided. How exactly was the question phrased? Could intake mean as little as a single bite or as much as a large meal? How much do parents really know about eating frequency at the older ages - is this at all realistic to ask them? Perhaps a considerable component of the apparent reduced meal frequency is because parents simply don't know when their older children are eating? It was also only a single day of measurement at each point. These limitations all need to be addressed more thoroughly, including comments that what was eaten was not measured in any way.6. Page 9 - says ethical approval was obtained in 2006 yet children were recruited for baseline at 2002?
---

REVIEWER	Mattea Dallacker Max Planck Institute for Human Development
-----------------	--

GENERAL COMMENTS

The longitudinal study investigates the association between meal frequency and BMI in children. The study found that children who eat less than 5 times gain more weight compared to children who eat more than 5 times a day. Eating behavior in low and middle income countries is an important topic and one strength of the study is the longitudinal design. My main concern with the study is the lack of theory and the insufficient definition of the mean concept. There is no explicit definition of "meal", no hypotheses and no theoretical background. Most of the cited studies in the introduction and discussion refer to the benefits of family meals, however, the present study measured "consumption of any food". It appears that the study outcome does not differentiate between snacking, eating on the go and a proper sit-down meal, although these eating occasions have very different health implications. Therefore, I think it is difficult to interpret the findings of the study and to draw any conclusions.

My comments are detailed below.

p. 6 ll. 20 ff.

This sentence is confusing. I suggest to change the sentence into: "Changes in BMI are explained by [...]. Therefore, some diet characteristics, such as [...]."

p.6 ll.24 ff.

What is meant by „meal frequency“? Please define what constitutes a meal and why it is healthy. Most of the cited studies, investigated family meals, as it has been shown that meals eaten together with the family are healthier compared to meals eaten alone.

p.6 ll. 29 ff.

Not clear which target group is addressed. Children or adults?

p.6 l. 43

The authors say that there is no study from Latin America having investigated the link between meal frequency and BMI. I agree that we need more diversity and a sample from Peru is an important contribution to the literature. However, please note, that there is at least one study by Serrano et al. (2014; also included in the meta-analysis by Dallacker et al) showing a negative association between meal frequency and BMI in Puerto Rican (i.e. Latin American) households. This should be acknowledged by the authors.

p.6, ll. 48

I agree, that we need more studies on family meals in middle or low income countries. The sentence "[...] there could be differences because of nutritional profiles and access to food patterns of these populations." is very vague and needs some explanation. It would be helpful if the authors elaborate more on

	this point and give few examples for what is meant by different nutritional profiles. p. 6 ll. 55 ff. Do the authors have any hypotheses? If not, it should be clearly stated that this is an explanatory study. p. 7, ll. 23 ff. Please give reasons for why older children were excluded from the analyses. p.8, ll. 19 ff. How was BMI measured? Please state whether BMI was measured or self-reported. If weight and height were self-reported, this should be mentioned in the limitation section. p. 8, ll. 41 The description of the scale to assess meal frequency is difficult to understand. It would be helpful if the authors give an example of the items that were used to assess meal frequency. Now, it reads as if the authors have measured the number of times the child has eaten food. Does the scale differentiate between snacking, eating on the go and a proper sit-down meal? This is important, because snacking and eating on the go have different health implications compared to proper sit down meals. Again, it remains unclear what is meant by “meal frequency”. p. 11, ll. 29 ff. The sentence “Less frequent meals were associated with lower BMI gain” contradicts the results. Please correct. p.11, l. 50 The systematic review by Dallacker and colleagues was conducted in 2018, not 2017. Please correct.
--	--

VERSION 1 – AUTHOR RESPONSE

Reviewer 1:

The authors present an interesting analysis of meal frequency in relation to BMI over time in young Peruvian children. Although this question has been examined previously many times, the authors quite rightly highlight that much of the existing evidence is cross-sectional in nature, and in children from high-income countries. I suggest some relatively small edits to the manuscript to improve the clarity of the data:

1. *It would perhaps be more useful to have the mean age of participants at each time point rather than the years of data collection.*

We agree with the reviewer regarding mean age is needed in the manuscript; however, we believe years of data collection should be used in the methods section. Nevertheless, we have included the information of mean age and SD in Table 2 as requested in a subsequent comment.

2. **Page 6, line 13 - reword to illustrate this more clearly - do the authors refer to the median over different ages? It is not entirely clear what this sentence is referring to.**

We have rephrased this sentence as requested. Now it reads: "As a result, information of the second round was used as the baseline of the cohort, and information from the third, fourth and fifth rounds were included as follow-up assessments."

3. **The mean (SD) age at each follow-up needs to be included - perhaps in Table 2? This would make the BMI data more useful - at present BMI itself is not particularly useful without knowledge of the relevant age of the children, and how variable that age was since BMI changes so rapidly across growth.**

As pointed out in comment 1, we have decided to add the mean age and SD in Table 2 according to reviewer's comment.

4. **What reference data were used to calculate the BMI z-scores?**

The WHO reference data was used to estimate BMI z-scores. We have added this information in the Methods section.

5. **More information - and acknowledgement of the limitations - regarding the meal frequency measure needs to be provided. How exactly was the question phrased? Could intake mean as little as a single bite or as much as a large meal? How much do parents really know about eating frequency at the older ages - is this at all realistic to ask them? Perhaps a considerable component of the apparent reduced meal frequency is because parents simply don't know when their older children are eating? It was also only a single day of measurement at each point. These limitations all need to be addressed more thoroughly, including comments that what was eaten was not measured in any way.**

We have added information about how the meal frequency variable was measured in the Methods section. Now it reads:

"The exposure of interest was meal frequency as reported by the parents based on seven questions related to the consumption of any possible meal or snack during the previous 24 hours before the interview. The question was asked as follow: "During the previous 24-hour period, did the child consume: any food before a morning meal (breakfast)? Any morning meal (breakfast)? Any food between morning and midday meals? Any midday meal? Any food between midday and evening meals? Any evening meal? Any food after the evening meal? A simple addition of yes/no responses to these seven questions was used to estimate the total number of meals the child ate in the last 24 hours".

In addition, we have added limitations regarding this point in the Discussion section. Now it reads: "First, parents' reports were used to evaluate the number of meals per day based on the last day before the interview; recall bias may arise as a concern. In addition, parents may not be aware of eating frequency of children, especially at older ages. Moreover, although questions assessed any meal or snack, portion (i.e. a large meal or a single bite) was not considered and, for instance, not evaluated as part of the analysis."

6. **Page 9 - says ethical approval was obtained in 2006 yet children were recruited for baseline at 2002?**

Sorry for the typo. We have corrected this.

Reviewer 2:

The longitudinal study investigates the association between meal frequency and BMI in children.

The study found that children who eat less than 5 times gain more weight compared to children who eat more than 5 times a day. Eating behavior in low and middle income countries is an important topic and one strength of the study is the longitudinal design. My main concern with the study is the lack of theory and the insufficient definition of the mean concept. There is no explicit definition of “meal”, no hypotheses and no theoretical background. Most of the cited studies in the introduction and discussion refer to the benefits of family meals, however, the present study measured “consumption of any food”. It appears that the study outcome does not differentiate between snacking, eating on the go and a proper sit-down meal, although these eating occasions have very different health implications. Therefore, I think it is difficult to interpret the findings of the study and to draw any conclusions.

p. 6 ll. 20 ff: This sentence is confusing. I suggest to change the sentence into: “Changes in BMI are explained by [...]. Therefore, some diet characteristics, such as [...].”

We have rephrased this sentence as recommended by the reviewer.

p.6 ll.24 ff: What is meant by „meal frequency“? Please define what constitutes a meal and why it is healthy. Most of the cited studies, investigated family meals, as it has been shown that meals eaten together with the family are healthier compared to meals eaten alone.

Based on the Young Lives handbooks and the questionnaire applied during the assessments, the seven questions evaluated the consumption of any food (questions asked about each possible meal or snack) as follows:

“During the previous 24-hour period, did the child consume:

- Any food before a morning meal (breakfast)?
- Any morning meal (breakfast)?
- Any food between morning and midday meals?
- Any midday meal?
- Any food between midday and evening meals?
- An evening meal?
- Any food after the evening meal?”

All the questions have only two options of response: yes or no, although in some cases a “Do not know” could be recorded. Based on that, we could not assess if the snack or meal was healthy or not. This information has been added in the text of the manuscript in Methods as well in Limitations section (See response to comment 5 of reviewer 1).

p.6 ll. 29 ff: Not clear which target group is addressed. Children or adults?

Children, when they were aged 4 to 5 years old at baseline, were evaluated. At the end of the follow-up, these children were adolescents (about 14 years old). This information has been included in the Table 2 as suggested by a previous reviewer.

p.6 l. 43: The authors say that there is no study from Latin America having investigated the link between meal frequency and BMI. I agree that we need more diversity and a sample from Peru is an important contribution to the literature. However, please note, that there is at least one study by Serrano et al. (2014; also included in the meta-analysis by Dallacker et al) showing a

negative association between meal frequency and BMI in Puerto Rican (i.e. Latin American) households. This should be acknowledged by the authors.

We have changed this information in our manuscript text to include the study by Serrano et al and the appropriate reference.

p.6, ll. 48: I agree that we need more studies on family meals in middle or low income countries. The sentence “[...] there could be differences because of nutritional profiles and access to food patterns of these populations.” is very vague and needs some explanation. It would be helpful if the authors elaborate more on this point and give few examples for what is meant by different nutritional profiles.

We have added a couple of lines regarding this point. Now it reads: “Therefore, whether the association between meal frequency and BMI holds in Latin America, is unknown; though there could be differences because of the nutritional profiles and access to food patterns of these populations. For example, food insecurity is high in Peru, especially in rural and poor areas (13), and households with an insecure profile usually have a poor diet quality and consume more carbohydrates (14).”

p. 6 ll. 55 ff: Do the authors have any hypotheses? If not, it should be clearly stated that this is an explanatory study.

We have added the hypothesis at the end of the Introduction as requested. Now it reads: “We hypothesized that children eating less frequently will have a greater gain of BMI over time.”

p. 7, ll. 23 ff: Please give reasons for why older children were excluded from the analyses.

We did not exclude older children from the analysis. There are two independent cohorts following different group of children; we analyzed only the younger cohort due to data availability.

p.8, ll. 19 ff: How was BMI measured? Please state whether BMI was measured or self-reported. If weight and height were self-reported, this should be mentioned in the limitation section.

Both weight and height were measured during each of the assessment of the children. They were not self-reported. We have included this information in the text.

p. 8, ll. 41: The description of the scale to assess meal frequency is difficult to understand. It would be helpful if the authors give an example of the items that were used to assess meal frequency. Now, it reads as if the authors have measured the number of times the child has eaten food. Does the scale differentiate between snacking, eating on the go and a proper sit-down meal? This is important, because snacking and eating on the go have different health implications compared to proper sit down meals. Again, it remains unclear what is meant by “meal frequency”.

We have clarified this point as requested in a previous comment.

p. 11, ll. 29 ff: The sentence “Less frequent meals were associated with lower BMI gain” contradicts the results. Please correct.

We have corrected this.

p.11, l. 50: The systematic review by Dallacker and colleagues was conducted in 2018, not 2017. Please correct.

This typo was corrected.

VERSION 2 – REVIEW

REVIEWER	Professor Rachael Taylor University of Otago New Zealand
REVIEW RETURNED	11-Mar-2020

GENERAL COMMENTS	The authors have addressed all previous queries to my satisfaction.
---

REVIEWER	Mattea Dallacker Max Planck Institute for Human Development
REVIEW RETURNED	27-Mar-2020

GENERAL COMMENTS	Now, in the title the term „food frequency“ instead of „meal frequency“ is used. I agree that „food frequency“ is a better description of the main outcome compared to “meal frequency”. The outcome was measured by the question: “During the previous 24-hour period, did the child consume: any food [...]?”. “Consuming food” is not equal to having a meal, but includes all kinds of eating occasions, such as snacking, taking one bite of something. Thus, I suggest to use the term food frequency instead of meal frequency throughout the manuscript.
--

VERSION 2 – AUTHOR RESPONSE

Reviewers' comments to Author:

Reviewer 1:

The authors have addressed all previous queries to my satisfaction.

Thanks.

Reviewer 2:

Now, in the title the term „food frequency“ instead of „meal frequency“ is used. I agree that „food frequency“ is a better description of the main outcome compared to “meal frequency”. The outcome was measured by the question: “During the previous 24-hour period, did the child consume: any food [...]?”. “Consuming food” is not equal to having a meal, but includes all kinds of eating occasions, such as snacking, taking one bite of something. Thus, I suggest to use the term food frequency instead of meal frequency throughout the manuscript.

We agree and we have done the changes as suggested.